# Improved drug-screening tests of candidate anti-cancer drugs in patient-derived xenografts through use of numerous measures of tumor growth determined in multiple independent laboratories

**Elizabeth Rosenzweig**[1], **David E. Axelrod**[2], **Derek Gordon**[3]*

**1** Department of Genetics and Institute of Quantitative Biomedicine, Rutgers-The State University of New Jersey, Piscataway, New Jersey, United States of America, **2** Department of Genetics, Human Genetics Institute, and Cancer Institute of New Jersey, Rutgers-The State University of New Jersey, Piscataway, New Jersey, United States of America, **3** Department of Genetics, Human Genetics Institute, Rutgers-The State University of New Jersey, Piscataway, New Jersey, United States of America

* derekgo@hginj.rutgers.edu

## Abstract

### Background

Researchers screen candidate anti-cancer drugs for their ability to inhibit tumor growth in patient-derived xenografts (PDXs). Typically, a single laboratory will use a single measure of tumor growth.

### Purpose

An effective drug-screening test as one that correctly identifies whether a drug treatment inhibits or does not inhibit tumor growth. We document improvements in the experimental design and statistical analysis of drug-screening tests based on the criteria of sensitivity and specificity.

### Methods

We analyzed two published datasets. The response of each PDX model was known in advance. This information provided for statistical ground-truth classification. One dataset analyzed growth inhibition in the presence of one specific drug treatment for two PDX tumor models for numerous labs. A second dataset reported tumor growth of many PDX models in the presence of many drugs. A PDX model for which the treatment showed no tumor growth inhibition is referred to as Progressive Disease (PD). A PDX model for which the treatment showed complete tumor growth inhibition is referred to as Completely Responsive (CR). We created and analyzed four drug-screening tests, based on p-values for either a single-measure and single-lab,

**Data availability statement:** All relevant data are within the paper and its Supporting Information files.

**Funding:** The author(s) received no specific funding for this work.

**Competing interests:** The authors have declared that no competing interests exist.

or p-values from meta-analysis and multiple-test correction. The outcome of each screening test was that either the drug treatment was effective or it was not. For both datasets, we computed median sensitivities and specificities by applying bootstrap resampling, and specification of a significance level.

## Results

Our results showed that drug screening tests utilizing p-values from meta-analysis of numerous labs, or multiple test correction, produced median sensitivities and specificities that were always at least as high as those for the Single-Measure, Single-Lab test. This result was true for all significance levels. The 95% confidence intervals were usually greater in length for the Single-Measure, Single-Lab screening test.

## Introduction

We develop methods for evaluating the results of preclinical drug-screening tests. We compare several different measures of drug inhibition of tumor growth in patient-derived xenografts (PDX), and evaluate the benefit of determining inhibition in different laboratories. The criteria of sensitivity, specificity, and accuracy determine the best experimental design and statistical analysis. The significance of this work is that more accurate drug-screening tests improve decision-making in selecting effective cancer treatments.

Determining the sensitivity, specificity, and accuracy of a drug-screening test requires that the test results are compared to results with known true positive and true negative outcomes, in this case a result in which there is complete inhibition of tumor growth or no inhibition of tumor growth [1]. We refer to this known true condition as ground truth or gold standard classification. However, the outcome is not known for novel compounds. Further, many drugs only partially inhibit the growth of many human-derived tumors, which reduces drug-screening test power. This presents a challenge when evaluating the sensitivity, specificity, and accuracy of the results of a drug-screening test on a new compound. The challenge can be overcome by using tumors derived from stable cell lines with reproducible response to specific drugs, either completely growth-inhibited or not [2].The drug-screening pipeline occurs in several steps. The first two steps are *in vitro*, to evaluate the ability of a library of molecules to inhibit the biochemical reaction of a target disease process; then *in vivo*, to determine which of the effective molecules in the first step inhibit the growth or survival of well-characterized cell lines in culture [3]. The third step tests the effective molecules from the second step in a more complex model, grafted tumors in mice. The tumors may be cell line derived xenografts (CDX) [4] or patient-derived xenografts (PDX) [5]. Both CDX and PDX have advantages and limitations [6,7].

Testing newly developed drugs for tumor inhibition in animals can be an informative step before human clinical trials [8]. There have been at least twenty-six recent reports of new anticancer drugs. Most reports have characterized the drugs with

biochemical tests, and some by inhibition of tumor cell growth *in vitro*. Most relevant to this article are the reports that also include inhibition of tumor growth in animals.

Zhu *et al.* [9] synthesized novel β-elemene nitric oxide donor derivatives for treating myeloid leukemia. They tested the derivatives for inhibition of the growth of several cell lines in culture, including human myeloid leukemia K562 cells. One compound, 18f, inhibited the growth rate of K562 cell mouse xenografts by 73.18%, as determined by the measurement of tumor volume at 29 days. Zhang *et al.* [10] developed a pH and reactive oxygen species (ROS) dual stimulus-responsive drug delivery system (PN@GPB-PEG NPs) loaded with chemotherapeutic paclitaxel (PTX) and indoleamine 2.3-dioxygenase (IDO) inhibitor NLG919. They determined a decrease in viability of hepatocellular carcinoma Hepa 1–6 cells in culture after 48 hours of treatment. The growth of Hepa 1–6 cell tumors in mouse xenografts were significantly inhibited, as determined by tumor volume measured after 15 days of treatment. Lin *et al.* [11] silenced OP18/stathmin by RNA interference, and in combination with taxol, demonstrated the inhibition of nasopharyngeal carcinoma cells (NPC) growing *in vitro*. They also determined the inhibition of NPC xenografted tumors at 40 days of treatment. Wu *et al.* [12] demonstrated that dimethyl fumarate (DMF) enhanced angstrom-scale silver particles (F-AgÅPs) successfully induce cytotoxicity of U266 multiple myeloma cells grown *in vitro*. They also determined that F-AgÅPs and DMF synergistically inhibited U266 cell tumor growth in xenografts, as measured by tumor volume over 11 days. Fu *et al.* [13] designed and synthesized derivatives of the oncolytic peptide LTX-315. The derivative FXY-12 inhibited the growth of cells in culture, including cancer cells that grow in suspension (A20, U937, and COC1), and adhesive cancer cell lines (HeLa, B16–F10, ES-2, MCF-7/ADR, and A549/T). The peptide FXY-12 also inhibited the growth of A20 cell line-derived mouse xenografts, determined by measurements of tumor volume over 22 days.

Each of the above publications reports a single statistical measure of tumor growth inhibition in animals, determined in a single laboratory. We refer to this approach as the Single Measure, Single Lab drug-screening test. In this work, we compare the sensitivity and specificity of this approach to approaches that consider results of numerous tumor growth measures from numerous laboratories.

Candidate anti-cancer drugs have been tested in animal models [14]. A typical experimental protocol is to implant a patient-derived tumor or implant a cell-line as a xenograft (PDX or CDX) into each of a group of five to ten mice and measure the tumor volume every three or four days for two or three weeks. An example experimental protocol is illustrated in Fig 1 below. Whether or not the drug is inhibitory, and the extent to which it may be inhibitory, is determined by comparing

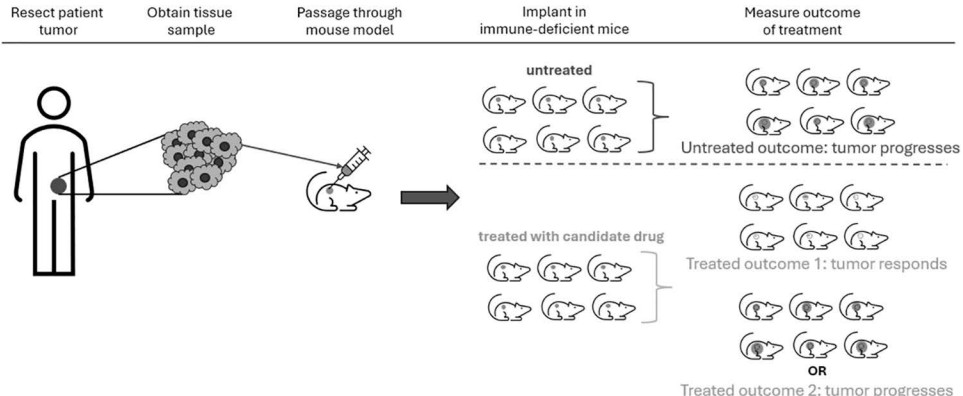

**Fig 1. Patient-derived xenograft experimental protocol.** Clinicians resect the patient tumor and passage it through mice to develop a PDX tumor model. The tumor model is implanted in several immunocompromised mice, which are then given either a candidate drug or a placebo. Researchers measure the tumor volume over time and use the change to determine whether the candidate drug inhibited tumor growth. Motivated by Fig 1 in Karamboulas *et al.* [15].

the growth curve of tumors treated with a drug and control tumors not treated. Publications have reported comparisons by one of several criteria.

A commonly reported heuristic is simply to ask if the drug is inhibitory, with a yes or no answer [14,16–22]. There are a wide variety of methods that can be applied to achieve a statistically rigorous analysis for such data. Some collections of those methods are presented in the Methods and Discussion sections below. Previous work by Gordon and Axelrod examined a statistical method involving grouping via PROC TRAJ for classifying drug efficacy in single mouse trials [23].

We apply bootstrap resampling to explore the sampling distribution of the data sets we used in comparing robustness between the standard Single-Measure, Single-Lab (SMSL) drug-screening test and our novel drug-screening tests. Resampling permits us to make comparisons among screening tests based on sensitivity, specificity, and accuracy.

One key aspect of this work is the consideration not only of numerous measures of tumor growth calculated on the same population of tumor growth curves but also of multiple testing corrections on those calculations. This correction allows us to make use of multiple data sets generated by a single PDX trial. Another aspect is the use of statistical meta-analysis methods to derive an "analysis of analyses" that can strengthen the conclusions drawn from data generated in numerous animal studies.

Our goal is to answer the question: Is it possible to formulate a drug-screening test that improves sensitivity and specificity compared with the Single-Measure, Single-Lab (SMSL) test? We answer this question by performing analyses on two real-world PDX data sets.

## Methods

### Ethics statement on human participants

None of the data used in this study came from human participants. No ethics committee approval was necessary because we did not conduct animal studies, nor did we use data from prospective or retrospective human research studies.

### Flow chart

We present a flow chart in Fig 2 to summarize some of the steps involved in creating and evaluating the novel drug-screening tests, in an effort to make the methods and results easier to follow.

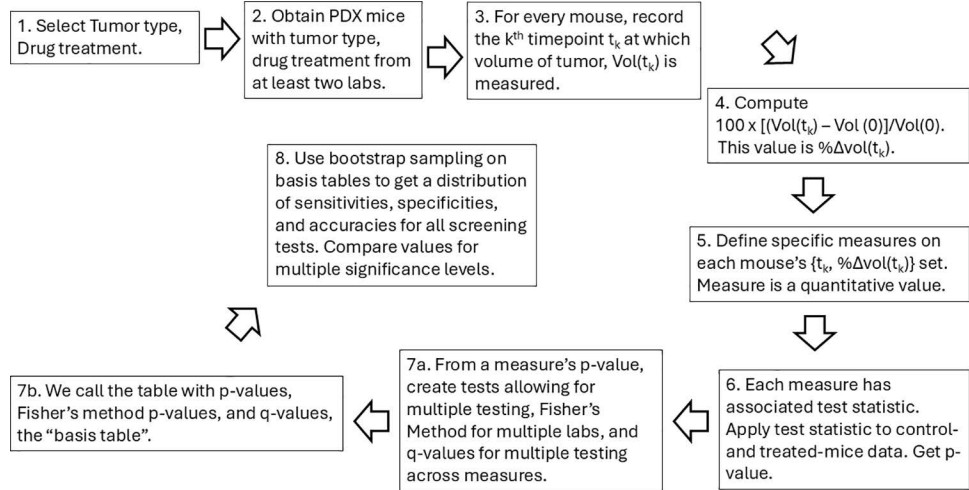

**Fig 2. Flowchart.** This chart summarizes the steps needed to create screening tests and bootstrap samples for generating distributions of each screening test at multiple significance levels.

Throughout this work, we will indicate places corresponding to the various steps of the flow chart. In S1 File, we perform all the steps in Fig 2 for an actual data set.

## Description of data sets

Using RECIST 1.1 [24] nomenclature, we define the positive screening test condition as completely responsive. This means the volume of the tumor decreases over a set number of days. We specify 21 days, a commonly used duration in PDX-trial studies [18,20,25]. We define the negative screening test condition as Progressive Disease (PD). This means the volume of the tumor increases over the 21 days of the study. We additionally distinguish between the actual condition or "ground truth classification" of the tumor response and the drug-screening test classification.

We used two data sets with different properties to develop the novel screening tests. The first, derived from the data used in the Evrard et al. paper [2] and kindly provided to us by Dr. Mike Lloyd at the Jackson Laboratory, contains one tumor model (TM) with positive ground truth in the presence of the candidate drug temozolomide, and one TM with negative ground truth. The positive ground truth TM is an engineered bladder sarcomatoid transitional cell carcinoma developed by the Jackson Laboratory for Genomic Medicine and identified by the NIH Patient-Derived Models Repository (PDMR) as BL0293-F563. The negative ground truth TM is a colon adenocarcinoma identified by PDMR as 625472–104-T. (Fig 2, Step 1). The ability of the drug temozolomide to inhibit the growth of these tumor models was determined in numerous laboratories, including the National Cancer Institute Patient-Derived Models Repository (PDMR), Huntsman Cancer Institute/Baylor College of Medicine (HCI-BCM), MD Anderson Cancer Center (MDACC), Washington University in St. Louis (WUSTL), and The Wistar Institute/University of Pennsylvania (WIST) (Fig 2, Step 2). The subset of the data we used is included in S2 and S3 Tables.

The second data set was derived from supplemental information published with the Gao et al. paper [5]. These data consisted of 2 replicates each of 35 individual breast cancer (BRCA) TMs with unknown ground truth classification in the presence of the candidate drug paclitaxel (Fig 2, Step 1). One replicate, which we call "treatment", received treatment with paclitaxel; the other replicate, which we call "control", received no treatment with a cancer drug. The data set is publicly available to download via the link in the References section.

## Threshold selection on BRCA-paclitaxel data

Since evaluating sensitivity and specificity for a screening test requires knowledge of both ground truth classification and drug-screening test classification, we established a proxy for ground truth classification on the Gao et al. data using threshold selection. We calculated two values defined in the "Measures" section — the area under the curve for the DTV at all time points (AUCmax) and the $DTV_{21}$. We sorted all 70 TMs from largest to smallest on both measures. We applied symmetric threshold selection [26,27], specifying those TMs in the upper 20% for both measures as having negative ground truth classification, and those TMs in the lower 20% as having positive ground truth classification. Of the TMs that fell within the threshold selection, 27 of 28 were correctly classified according to their treatment status as indicated by Gao et al. One TM that was threshold-selected as having negative ground truth classification belonged to the "treatment" cohort of the Gao et al. data set. This corresponds to a 96% correct classification rate.

For the Gao et al. data, specifically the CR set, we generated five simulated data sets, each one generated by randomly selecting (with replacement) 12 treated mice and 11 control mice (difference in mouse/TM numbers due to total number of ground-truth mice (threshold selection)). We performed simulations because the Gao et al. data only have two mice/TM for the drug treatment. Said another way, Gao et al. has only one lab's worth of data, whereas Evrard et al. has five labs' worth of data. We repeated the simulation process five times to generate five simulated labs worth of data. These data are available in S4 and S5 Tables.

For the PD set, we applied the same simulation process, with the exception that the "treated" mice were drawn from negative ground truth classification as well as the control mice. All other numbers are the same (Fig 2, Step 2).

Similarities among the Evrard *et al.* and Gao *et al.* data sets are that they both considered PDX trials, both had candidate drug treatments, the data generated from the tumor models and drug treatments consisted of longitudinal tumor growth data, specifically the measure of tumor size in each mouse at given time points. Because of this experimental design, we were able to apply the same statistical tests to data from each set, and subsequently, compute the outcomes for each drug screening test.

## Statistical measures used

Central to all calculations in this work are the values Vol(t) and %ΔVol(t). For a given mouse, Vol(t) is equal to the raw tumor volume of the tumor model at day t. The percent relative change in tumor volume at time t (in days), %ΔVol(t), is arguably the most important value in this work. All measures (presented below) are defined using this value. It is:

$$\%\Delta Vol(t) = 100 \times \frac{(Vol(t) - Vol(0))}{Vol(0)}.$$

For each mouse, the set of coordinates $\{(t_k, \%\Delta Vol(t_k)\}$, where $\%\Delta Vol(t_k)$ exists at times $t_k$ for a given mouse, is referred to as the empirical tumor growth trajectory (ETGT) (Fig 2, Steps 3 and 4).

## 95% confidence interval for the bootstrap distribution

The lower endpoint (lower confidence limit, or LCL) is the 2.5th percentile of the bootstrap distribution, and the upper endpoint (upper confidence limit, or UCL) is the 97.5th percentile. We may use the notation 95%CI [LCL,UCL], when LCL and UCL are known. The length of the confidence interval is defined as (Upper Confidence Limit – Lower Confidence Limit).

In Table 1, we provide a concrete example of Vol(t) with calculated %ΔVol(t). The example tumor volumes Vol(t) were recorded on the $k^{th}$ day (with corresponding day indicated by each column). The relative change in a given day's Vol(t) value was computed using the percent relative difference in tumor volume formula from the notation above (Fig 2, Step 3).

## Measures

We define each measure in more detail as follows; that is, the quantitative value for a given mouse for t days (Fig 2, Step 5).

**Measure 01: DTV$_t$** = %ΔVol(t). In this work, we focus on day t = 21, as it is customarily studied, and most experimental designs have measurements going out to at least 21 days.

The corresponding test statistic is the Welch's t-statistic on the groups (control and treated), where the quantitative measure is **DTV$_t$**. Note that this measure is only defined for mice who have a pair of coordinates (t, %ΔVol(t)), where t ≥ 21 days.

For Measures 02 and 03, TotalDays is the total number of days for which %ΔVol(t) is measured. As an example, in Table 1, TotalDays = 7. If we were to remove the last two columns, then TotalDays = 5.

**Table 1. An example of data for the Empirical Tumor Growth Trajectory (ETGT).**

| Variable | $t_k$ day ($k^{th}$ day relative to start of experiment) | | | | | | |
|---|---|---|---|---|---|---|---|
| | 0 days (k=1) | 4 days (k=2) | 7 days (k=3) | 11 days (k=4) | 14 days (k=5) | 18 days (k=6) | 21 days (k=7) |
| Tumor Volume Vol($t_k$) | 138.90 | 157.70 | 231.80 | 326.10 | 393.60 | 649.00 | 702.30 |
| %ΔTV($t_k$) | 0.00 | 13.53 | 66.88 | 134.77 | 183.37 | 367.24 | 405.62 |

The days on which Vol(t) is measured are indexed by k, and the actual day on which the Vol(t) is measured is $t_k$. Also, the values %ΔVol(t) are computed for each day $t_k$. In Fig 3, we plot the ETGT $\{(t_k, \%\Delta Vol(t_k)\}$ for the mouse data in Table 1, 1 ≤ k ≤ 7.

**Measure 02: $AUC_t$ = Area under the curve for the tumor growth trajectory to time t**

$$\sum_{k=1}^{TotalDays-1} \left( \frac{1}{2} \times (t_{k+1} - t_k) \times (\%\Delta Vol(t_{k+1}) + \%\Delta Vol(t_k)) \right)$$

For those mice in which $\%\Delta Vol(t_k)$ exists. In this work, we set $t_{TotalDays}$ = 21 days. This formula is a sum of trapezoids, specifically the area under the curve for a ETGT. The corresponding test statistic is the two sample Welch's t-statistic, computed for the mean $AUC_{21}$ values on control mice versus the mean $AUC_{21}$ values on treated mice. This measure is only defined for mice who have a pair of coordinates (t, $\%\Delta Vol(t)$), where t ≥ 21 days.

**Measure 03: AUCmax = Area under the curve for the entire tumor growth trajectory**

$$\frac{1}{TotalDays} \times \sum_{k=1}^{TotalDays-1} \left( \frac{1}{2} \times (t_{k+1} - t_k) \times (\%\Delta Vol(t_{k+1}) + \%\Delta Vol(t_k)) \right)$$

The differences between this measure and $AUC_t$ are that a mouse need not have a value $\%\Delta Vol(t_k)$ at $t_k$ = 21 days and that the measure is normalized by the length of the trajectory for direct comparison of EGTGs of different durations. Again, $t_{TotalDays}$ ≤ 21 days. Like Measure 01, the corresponding test statistic is the two-sample Welch's t-statistic, computed for the mean AUCmax values on control mice versus the mean AUCmax values on treated mice. This measure is only defined for mice that have a minimum survival time of $t_{TotalDays}$ = 10 days.

**Measure 04: Tumor Growth Inhibition at time t = $TGI_t$ = $R_{t,j}$ = ln[($\%\Delta Vol(t)$/100) + 1]**, for a mouse in Group j. If the mouse is in the control group, we set j = 0. If the mouse is in the treatment group, we set j = 1. The term $R_{t,j}$ is the ratio of tumor size from baseline to time t and algebraically can be shown to be a linear equation in $\%\Delta Vol(t)$ (shown above). As with other measures, t = 21 days, so $R_{21,j}$ = $\%\Delta Vol(21)$/100 + 1.

The corresponding test statistic is the Ordinary Least Squares (OLS) regression for the model: $\ln(R_{21,j}) = \alpha + \beta T_j + \varepsilon_j$, $\varepsilon_j \sim N(0,\sigma^2)$. In the regression formula, $T_j$ is the indicator function for whether a mouse is in the control group ($T_j = 0$), or in the treatment group ($T_j = 1$). The coefficients α and β are estimated from the OLS regression. Because there are only two groups, the F distribution of the test statistic is mathematically equivalent to Student's t distribution.

The null hypothesis is H0: β = 0, that is, $\%\Delta Vol(21)$ does not depend upon treatment status.

**Measure 05: $PFS_\delta$ = Progression-free survival for δ-fold increase in $\%\Delta Vol(t)$**. There are two components to this measure:

1. The day t at which a mouse's $\%\Delta Vol(t)$ value exceeds δ × 100.

2. The censoring status.

We set t = 21 days. If a ETGT's $\%\Delta Vol(t)$ value exceeds δ × 100 at any day up to and including t = 21 days, we state that the ETGT has not been censored and set the censoring indicator function to 0.

If a mouse's $\%\Delta Vol(t)$ does not exceed δ × 100 for all days up to and including t = 21 days, we state that the ETGT has been right-censored and set the censoring indicator to 1. In this instance, the censoring day is t = 21.

The statistic for the $PFS_\delta$ measure is the log-rank statistic applied to the control and treatment groups, with each mouse's data being the pair of values indicated above (day, censoring value). It can be represented as a contingency table in which each row up to the last represents a day at which a progression event was recorded and the number of

progression events observed on that day, and the last row represents the number of ETGT's for which a progression event was not observed and the trajectory is right-censored.

The null hypothesis for this measure is that the counts of progression events in the contingency table are the same for the control and treatment cohorts. This can also be expressed as the hypothesis that the hazard functions are identical.

### Test statistics corresponding to each measure

Above, we indicated the corresponding test statistic for each measure. We summarize these results in Table 2. For each test statistic there is a null distribution. It is this distribution that allows us to compute p-values. Also, the test statistic for each tumor growth measure is listed in that table. Example calculations of all statistics may be found in the S1 File.

### The basis table

The basis table, Table 3, consists of all measure and lab p-values and corresponding q-values in a PDX trial analysis. The q-values are involved in multiple testing correction; see section "Using Storey's q-value for false discovery rate multiple testing correction on numerous measures" below. The p-values are determined from the test statistics in Table 2 (Fig 2, Step 6). Our basis table contains all the information we need to compute the outcomes of the screening tests. All notation is described in the text.

**Table 2. Summary of tumor growth measures, input for test statistics, and actual test statistics.**

| Measure (per mouse) | Statistic value | Test statistic |
|---|---|---|
| $DTV_t$ | Mean(control $DTV_t$) – Mean(treatment $DTV_t$) | Two-sample Welch test |
| $AUC_t$ | Mean(control $AUC_t$) – Mean(treatment $AUC_t$) | Two-sample Welch test |
| AUCmax | Mean(control AUCmax) – Mean(treatment AUCmax) | Two-sample Welch test |
| $TGI_t$ | Regression of $\ln(R_{t,j})$ on groups $j=0$ (control) and $j=1$ (treated) mice. | F-test (equivalent to t-test) |
| $PFS_\delta$ | Survival curves. Event defined as shortest day t that %ΔVol(t)> $\delta \times 100$ for a given mouse. | Log-rank test |

%ΔVol(t): The percent relative change from day $t=0$ of the tumor growth at day t.

$DTV_t$: %ΔVol(t).

$AUC_t$: Area under the curve for the tumor growth trajectory to time t.

AUCmax: Area under the curve for the entire tumor growth trajectory.

$TGI_t$: Tumor Growth Inhibition at time t, $\ln[(\%\Delta Vol(t)/100) + 1]$.

$PFS_\delta$: Progression-free survival for δ-fold increase in %ΔVol(t).

**Table 3. Basis table using measure p-values; measure q-values computed from p-values using Storey's q-value method.**

| p- and q-value rows | Measure | Lab | | | | | Fisher's Method |
|---|---|---|---|---|---|---|---|
| | | Lab 1 | Lab 2 | … | Lab (L-1) | Lab L | |
| p-values | Measure 1 | $p_{(1,1)}$ | $p_{(1,2)}$ | … | $p_{(1,L-1)}$ | $p_{(1,L)}$ | $p_{(1,+)}$ |
| | Measure 2 | $p_{(2,1)}$ | $p_{(2,2)}$ | … | $p_{(2,L-1)}$ | $p_{(2,L)}$ | $p_{(2,+)}$ |
| | … | … | … | ... | | … | … |
| | Measure (M-1) | $p_{(M-1,1)}$ | $p_{(M-1,2)}$ | … | $p_{(M-1,L-1)}$ | $p_{(M-1,L)}$ | $p_{(M-1,+)}$ |
| | Measure M | $p_{(M,1)}$ | $p_{(M,2)}$ | … | $p_{(M,L-1)}$ | $p_{(M,L)}$ | $p_{(M,+)}$ |
| q-values | Measure 1 | $q_{(1,1)}$ | $q_{(1,2)}$ | … | $q_{(1,L-1)}$ | $q_{(1,L)}$ | $q_{(1,+)}$ |
| | Measure 2 | $q_{(2,1)}$ | $q_{(2,2)}$ | … | $q_{(2,L-1)}$ | $q_{(2,L)}$ | $q_{(2,+)}$ |
| | … | … | … | ... | | … | … |
| | Measure (M-1) | $q_{(M-1,1)}$ | $q_{(M-1,2)}$ | … | $q_{(M-1,L-1)}$ | $q_{(M-1,L)}$ | $q_{(M-1,+)}$ |
| | Measure M | $q_{(M,1)}$ | $q_{(M,2)}$ | … | $q_{(M,L-1)}$ | $q_{(M,L)}$ | $q_{(M,+)}$ |

We give a fuller description of the correspondence of the values in this table with drug-screening tests in the section "Standard and Novel-screening tests."

## Meta-analysis of data from numerous labs with Fisher's method

Paraphrasing Yoon *et al.* [28], there is a high sensitivity, or ability to detect true positives, for meta-analysis methods like Fisher's method that combine a measure's p-values across numerous labs. This high sensitivity extends to situations where only a subgroup of the combined datasets have a nonzero effect size. Yoon *et al.* call this condition incomplete association. We consider incomplete association in meta-analysis in this section.

## Fisher's method for computing single p-value from numerous labs' p-values

We now demonstrate how to compute Fisher's Method statistic and corresponding p-value.

Consider the set of p-values for any measure, $1 \leq m \leq M$, and labs $1,\ldots,L$:

$$\left\{ p_{(m,1)}, p_{(m,2)}, \ldots, p_{(m,L)} \right\}.$$

Compute:

$$X^2 = (-2) \times \left[ \sum_{l=1}^{L} \ln\left( p_{(m,l)} \right) \right],$$

where ln is the natural-log function. Then $X^2$ is the Fisher's method statistic, and under the null hypothesis that the measure's p-values follow a uniform distribution for every lab, $X^2$ follows a central chi-square distribution with $2 \times L$ degrees of freedom. Rejection of the null suggests incomplete association, as defined by Yoon *et al.* above; specifically, at least one lab classifies the tumor type as CR, not PD, for the drug treatment.

## Using Storey's q-value for False Discovery Rate multiple testing correction on numerous measures

As has been documented in statistical and statistical-genetics methods [29–31], applying multiple statistical tests to the same set of data can increase the false positive rate. For numerous measures' p-values on a given lab, we determine a set of q-values using Storey's method, from which we can declare a result significant after correction for multiple testing.

Consider the following list of M p-values (one for each measure) for the $l^{th}$ lab, where l is fixed:

$$\left\{ p_{(1,l)}, p_{(2,l)}, \ldots, p_{(M,l)} \right\}.$$

Sort this list, so that:

$$p_{(M,l)} \geq p_{(M-1,l)} \geq \ldots \geq p_{(2,l)} \geq p_{(1,l)}.$$

Define the $j^{th}$ q-value as:

$$q_{(j,l)} = \begin{cases} p_{(M,l)}, & j = M, \\ \min\left( q_{(j+1,l)}, \frac{M}{j} p_{(j,l)} \right), & 1 \leq j < M \end{cases}.$$

We state that a q-value $q_{(j,l)}$ is significant at the $\alpha$ level after multiple-test correction with Storey's q-value method, or simply after multiple-test correction, if $q_{(j,l)} \leq \alpha$, where $\alpha$ is user-specified significance level. If this condition is met, we reject the null hypothesis that all p-values are drawn from a U[0,1] distribution.

## Standard and novel drug-screening tests

We employ a combination of multiple testing and meta-analysis to derive the novel screening tests SMNL, NMSL and NMNL, as summarized in Table 4.

For the screening tests below, we use the notation α to refer to the significance level.

The **Single-Measure, Single-Lab** (SMSL) screening test serves as the control to which we compare the novel screening tests. This screening test is the most common screening test used by researchers who perform PDX studies [16,32,33]. That is, many research teams apply a single measure's test statistic to a single lab's data for a given tumor type to establish whether the drug is inhibitory for the tumor type. The null hypothesis is that the p-value for the lab is drawn from a U[0,1] distribution, meaning the drug is not effective for the experiment done at that lab. The upper-left portion of the basis table (Table 3; clear cells) are the values used to compute the SMSL screening test outcomes.

**Single-Measure, Single-Lab decision rule for tumor type classification.** For a given p-value $p_{(m,l)}$, $1 \leq m \leq M$, $1 \leq l \leq L$ of the $M \times L$ clear cells in the basis table (upper left corner), we state that the tumor type is completely responsive to the drug treatment if $p_{(m,l)} \leq \alpha$, and is progressive disease if $p_{(m,l)} > \alpha$.

Next, we define the **Single-Measure, Numerous-Lab** (SMNL) screening test. For a fixed tumor type and measure m, $1 \leq m \leq M$, we combine all L labs' p-values into a single p-value using Fisher's method. The null hypothesis of the SMNL test is the p-values for each lab are independently drawn from a U[0,1] distribution; that is, the drug is non-inhibitory/tumor type is progressive disease for all of the labs.

**Single-Measure, Numerous-Lab decision rule for tumor type classification.** For a given measure m and corresponding p-value $p_{(m,+)}$, $1 \leq m \leq M$, (the M light gray cells in Table 3 (upper right corner)), we specify that the tumor type is completely responsive to the drug treatment if $p_{(m,+)} \leq \alpha$, and is progressive disease if $p_{(m,+)} > \alpha$.

A third drug-screening is the **Numerous-Measures, Single-Lab** (NMSL) screening test. For a chosen lab $l$ and tumor type, we have the M p-values $\{p_{(m,l)}, 1 \leq m \leq M\}$ (upper right corner of basis table) corresponding to the M measures for that lab. We apply false discovery rate correction with Storey's q-value method to obtain a vector of M q-values $\{q_{(m,l)}, 1 \leq m \leq M\}$. These values are directly below in the same column as the p-values (lower left corner of basis table; gray cells). The null hypothesis for this screening test is that no tumor growth measure has a positive drug-screening test classification across all measures considered in the test.

**Numerous-Measures, Single-Lab decision rule for tumor type classification.** For a given set of q-values $q_{(m,l)}$, $1 \leq m \leq M$, l fixed (gray cells in the basis table (lower left corner)) corresponding to the set of p-values, $p_{(m,l)}$, $1 \leq m \leq M$, (same l) in the same column, we state that the tumor type is completely responsive to the drug treatment if at least one of the M q-values satisfies $q_{(m,l)} \leq \alpha$. The tumor type is PD if no $q_{(m,l)} \leq \alpha$.

**Table 4. Definitions of novel screening tests.**

| Screening test | Multiple test correction | Null hypothesis Drug not inhibitory for: | Denominator for computing sensitivity or specificity |
|---|---|---|---|
| SMSL | None | specific measure and lab | $M \times L$ |
| SMNL | None | specific measure and all labs | $M$ |
| NMSL | q-values (M p-values) | all measures for a specific lab | $L$ |
| NMNL | q-values (M Fisher's p-values) | all measures and all labs | 1 |

SMSL: Single-Measure, Single-Lab.

SMNL: Single-Measure, Numerous-Labs.

NMSL: Numerous-Measures, Single-Lab.

NMNL: Numerous-Measures, Numerous-Labs.

Finally, the **Numerous-Measures, Numerous-Labs** (NMNL) screening test applies false discovery rate correction to the Fisher p-values (SMNL) for all five measures. The null hypothesis is that, for all measures, none of the Fisher p-values are significant at the $\alpha$ level after multiple testing correction. This statement means that the drug is not inhibitory for any lab or any measure.

**Numerous-Measures, Numerous Labs decision rule for tumor type classification.** For the set of set of q-values $q_{(m,+)}$, $1 \leq m \leq M$, for the column header "Fisher's Method" (steel blue cells in the basis table (lower right corner)) corresponding to the set of p-values, $p_{(m,+)}$, $1 \leq m \leq M$, we state that the tumor type is completely responsive to the drug treatment if at least one of the M q-values satisfies $q_{(m,+)} \leq \alpha$. The tumor type is progressive disease if no $q_{(m,+)} \leq \alpha$.

Each of our decision rules is designed to maintain a false positive rate equal to the significance level. That is, Pr(decision rule indicates that the tumor type is CR | ground-truth classification is that tumor type is PD) = α.

## Bootstrap method to calculate sensitivity and specificity confidence intervals

We evaluate the sensitivity and specificity of all drug screening tests to assess what tests have the highest values, and under what conditions. Yerushalmy [34] defines sensitivity as the probability of correct diagnosis of positive cases, and specificity as the probability of correct diagnosis of negative cases. He further discusses the challenge of assessing sensitivity and specificity "without reference to a standard" when one is not available.

For the Evrard *et al.* paper, there are known ground-truth classifications for each tumor type and the drug treatment, and therefore, we can directly calculate the sensitivity and specificity of any drug-screening test. For Gao *et al.*, the ground-truth classifications are determined by those TMs that have the same classification (CR or PD) by threshold selection on two independent methods, namely top- and bottom-20% of the sorted AUCmax and $DTV_{21}$ (measure) values for each PDX mouse.

Given Table 5's notation below, we define sensitivity as TP/(TP+FN) and specificity as TN/(TN+FP). The classification methods in bold (Tumor Model Ground Truth Classification in the third and fourth rows, Tumor Model Drug Screening Test Decision/Classification in the third and fourth columns) indicate how each cell in gray is computed. For example, if a tumor model is known to be CR to a drug treatment, i.e., the Tumor Model Ground Truth Classification is Completely Responsive, then TP is the count of all PDX mice that jointly have that tumor model and a drug-screening test decision of Completely Responsive. Similarly, if a tumor model is known to be PD to a drug treatment, i.e., Tumor Model Ground Truth Classification is Progressive Disease, then TN is the count of all PDX mice that jointly have that tumor model and a drug-screening test decision of Progressive Disease.

We can determine sensitivities and specificities for our actual data sets by determining the values in the basis table (Table 3). The theoretical distributions for the novel drug-screening tests are intractable to derive, so we cannot employ the standard methods for determining mean, median, and 95% significance levels. Such values indicate what screening tests may be optimal, e.g., highest mean, smallest confidence interval. As an alternative, we apply bootstrap resampling [35] using the actual data sets. With this approach, we can estimate parameters such as the median, and compute 95% confidence intervals.

To generate bootstrap resamples, we apply stratified resampling with replacement. Specifically, for a given group, a specific lab, and a specific tumor type, we create a bootstrap sample. We do this by replacing each mouse in a set with a

**Table 5. Ground-truth classification vs. drug screening test notation and decision.**

| | | Tumor Model Drug Screening Test Decision/ Classification | | |
|---|---|---|---|---|
| | | Completely Responsive | Progressive Disease | Total |
| **Tumor Model Ground-truth Classification** | Completely Responsive | True Positive (TP) | False Negative (FN) | TP + FN |
| | Progressive Disease | False Positive (FP) | True Negative (TN) | FP + TN |

randomly selected mouse from the same set. Note that, for some, the resampled mouse may be the same as the original mouse. We provide an example bootstrap sample in Table 6.

We generated 10,000 stratified bootstrap resamples for each of the Evrard *et al.* and Gao *et al.* CR data sets. We additionally created 10,000 bootstrap resamples each on the Evrard *et al.* and Gao *et al.* PD data sets, S6 Tables. For each bootstrap resample, we assembled the basis table, calculated the results of all four drug-screening tests (Table 4), and compared the drug-screening test classification of each screening test to the ground truth classification of the data.

For assessing sensitivity and specificity of the four drug screening tests, we examined the relationship between actual and drug-screening test classification at multiple significance levels commonly used in preclinical cancer drug and bioinformatics research. These significance levels, that we designate α, represent the approximate allowable type I error rate for each drug screening test in a theoretical experiment. Comparing the drug screening tests at a range of α including 0.1, 0.05, 0.01, and 0.001 allows us to assess the relative robustness of each test on sensitivity and specificity. The values 0.1 and 0.05, are commonly used and produce higher screening-test sensitivities. The values 0.01 and 0.001 were chosen to allow for more tests performed, such as in a high-throughput drug-screening assay. Using each of these two significance levels produces higher screening test specificities.

Stated another way, a more sensitive test will match ground truth classification positive to drug-screening test classification positive more often as α levels increase. Meanwhile, a more specific test will match ground truth classification negative to predictive condition negative as α levels decrease. A more accurate test (defined below) will be both more sensitive and more specific over the range of significance levels.

## Results

Drug screening tests for candidate anti-cancer drugs from a single laboratory may include multiple drugs and multiple tumors [5]. The reproducibility of drug screening tests can be determined by comparing the results of multiple statistical tests from numerous laboratories that treat the same tumors with the same drugs [2]. We perform these comparisons in the work that follows. The term precision refers to the length of the 95% confidence interval. The smaller the length, the greater the precision, and vice versa. Also, in all sections that follow, we employ the notation from Table 3, M = 5, L = 5 (Fig 2, Step 8).

### Sensitivity

**Evrard *et al.* data basis tables with novel drug screening test results for completely responsive data sets.** To determine the sensitivity of drug-screening tests, we ask how often different laboratories, using different statistical tests, classify as Completely Responsive a tumor that has a ground-truth classification of being completely responsive. Table 7

**Table 6. Sample illustration of stratified bootstrapping technique.**

| Replicate | ID | AUCmax | Mean | | Bootstrap ID | AUCmax | Mean |
|---|---|---|---|---|---|---|---|
| Control01 | 0 | 156.60 | 142.66 | | 2 | 107.44 | 171.46 |
| Control02 | 1 | 95.73 | | | 3 | 210.90 | |
| Control03 | 2 | 107.44 | | | 0 | 156.60 | |
| Control04 | 3 | 210.90 | | | 3 | 210.90 | |
| Treated01 | 4 | −60.25 | −53.17 | | 7 | −43.21 | −50.23 |
| Treated02 | 5 | −60.63 | | | 6 | −48.60 | |
| Treated03 | 6 | −48.60 | | | 4 | −60.52 | |
| Treated04 | 7 | −43.21 | | | 6 | −48.60 | |

AUCmax: Area under the curve for the entire tumor growth trajectory.

**Table 7. Basis table for the p-values and q-values from the Evrard *et al.* CR data set.**

| | Measure | Lab | | | | | Fisher's Method |
|---|---|---|---|---|---|---|---|
| | | HCI-BCM | MDACC | PDMR | WUSTL | WIST | |
| p-values | DTV$_{21}$ | 5.10E-04 | 6.50E-04 | 1.41E-08 | 9.00E-05 | **2.27E-03** | 2.38E-16 |
| | AUC$_{21}$ | 3.90E-04 | 8.20E-04 | 7.46E-08 | 6.10E-04 | **1.30E-03** | 3.69E-15 |
| | AUCmax | 1.50E-04 | 6.30E-04 | 4.57E-06 | 4.00E-05 | **1.62E-03** | 5.25E-15 |
| | TGI$_{21}$ | 2.15E-09 | **3.81E-03** | 1.06E-21 | 2.72E-10 | 4.04E-09 | 7.25E-44 |
| | PFS$_4$ | 2.00E-04 | **1.84E-03** | 1.99E-06 | 2.30E-03 | **2.33E-03** | 4.67E-13 |
| q-values | DTV$_{21}$ | 5.10E-04 | 1.37E-03 | 3.53E-08 | 1.50E-03 | 2.33E-03 | 5.94E-16 |
| | AUC$_{21}$ | 4.90E-04 | 1.37E-03 | 1.24E-07 | 7.60E-03 | 2.33E-03 | 6.11E-15 |
| | AUCmax | 3.30E-04 | 1.37E-03 | 4.57E-06 | 1.00E-03 | 2.33E-03 | 6.56E-15 |
| | TGI$_{21}$ | 1.08E-08 | 3.81E-03 | 5.30E-21 | 1.36E-09 | 2.02E-08 | 3.62E-43 |
| | PFS$_4$ | 3.30E-04 | 2.30E-04 | 2.48E-06 | 2.30E-03 | 2.33E-03 | 4.67E-13 |

The values in clear cells are the results of the 25 SMSL tests carried out. There are additionally 5 SMNL test results (light gray rows), 5 NMSL test results (dark gray columns), and 1 NMNL test result (steel blue column).

CR: Completely Responsive.

%ΔVol(t): The percent relative change from day t = 0 of the tumor growth at day t.

DTV$_t$: %ΔVol(t).

AUC$_t$: Area under the curve for the tumor growth trajectory to time t.

AUCmax: Area under the curve for the entire tumor growth trajectory.

TGI$_t$: Tumor Growth Inhibition at time t, $\ln[(\%\Delta Vol(t)/100)+1]$.

PFS$_\delta$: Progression-free survival for δ-fold increase in %ΔVol(t).

PDMR: Patient-Derived Models Repository of the National Cancer Institute.

HCI-BCM: Huntsman Cancer Institute/Baylor College of Medicine.

MDACC: MD Anderson Cancer Center.

WUSTL: Washington University in St. Louis.

WIST: The Wistar Institute/University of Pennsylvania.

SMSL: Single-Measure, Single-Lab.

SMNL: Single-Measure, Numerous-Lab.

NMSL: Numerous-Measures, Single-Lab.

NMNL: Numerous-Measures, Numerous-Labs.

is the basis table for the Evrard *et al.* CR data set. P-values for each cell of SMSL test using the appropriate statistical test for the corresponding measure of tumor growth ($p_{(m,l)}$ (Table 3) = p-value for the $m^{th}$ measure and $l^{th}$ lab; corresponding statistic for each cell's p-value comes from Table 2). For Fisher's Method p-values, we apply meta-analysis to the vector containing the p-values for all L laboratories for one tumor growth measure m to obtain a single meta-analysis p-value, $p_{(m,+)}$. For the single-laboratory l with five p-values {($p_{(m,l)}$, $1 \leq m \leq M$}, we compute five q-values {($q_{(m,l)}$, $1 \leq m \leq M$}, in the same column and directly below the p-values. We obtain these q-values through application of Storey's method. Finally, for Fisher's Method q-values, we apply the q-value multiple testing correction to Fisher's Method to obtain a set of five q-values, {($q_{(m,+)}$, $1 \leq m \leq M$} (Fig 2, Steps 6, 7a, and 7b).

The p-values in the basis table are a confirmation of the results published by Evrard *et al.* [2] In cases where the p-value was less than 0.001, we report the precise numerical result of the SMSL screening test. All four drug screening tests, including the SMSL test, have matching drug-screening test and ground truth classifications for the significance levels 0.1, 0.05, and 0.01. That means that the sensitivity of the four screening tests is 1.00 for these three significance levels. The SMSL test has a sensitivity less than 1.00 at α = 0.001. We highlight in bold those p-values greater than 0.001, a total of six p-values. Thus, the SMSL sensitivity of 18/25 = 0.76 at the 0.001 significance level. Notice that the five Fisher's

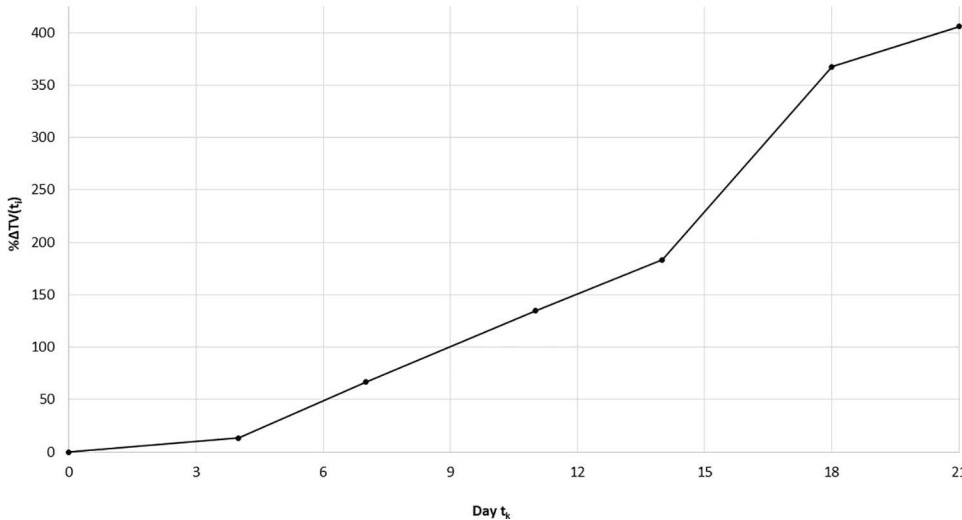

**Fig 3. Illustration of empirical tumor growth trajectory (ETGT) {(t_k, % ΔVol(t_k)}.** The %ΔVol(t) adjusted tumor growth normalizes TM growth measurements for statistical comparison of growth between TMs or replicates of the same TM.

method p-values are all less than 0.001, and the cells corresponding to each NMSL (dark gray columns) and NMNL (steel blue column) all have at least one value less than 0.001. That is, the sensitivity of the SMNL, NMSL, and NMNL screening tests are all 1.00 for α = 0.001.

**Median sensitivity and specificity for drug screening tests with 95% CIs.** The values in Table 7 are point estimates derived for a single data set. To obtain a distribution of screening test sensitivities, we apply the bootstrap sampling method. In Fig 4, we present results derived from performing 10,000 bootstrap samples with corresponding basis tables. We see that the three novel screening tests that use multiple measures, specifically NMSL, SMNL, and NMNL, have median sensitivities of 1.0 for all significance levels. The estimated sensitivity, with 95% confidence, for each of these drug-screening tests at all significance levels is also 1.0. The 95% confidence interval-lengths for the three novel screening tests are all 0, since the intervals are [1,1] for each screening test.

This result is not true for the SMSL screening test. The median sensitivities of the SMSL test are 1.0, 1.0, 0.96, and 0.8 for significance levels 0.10, 0.05, 0.01, and 0.001. The lengths of the 95% CIs for SMSL increase as the significance level decreases, with values 0.04, 0.04, 0.12, and 0.28 for the corresponding set of significance levels (Fig 4). This result suggests that the SMSL test is equally less powerful and less accurate than the novel drug screening tests.

**Gao *et al.* data basis tables with novel drug screening test results for completely responsive data sets.** Table 8 reports the numerical p- and q-values for the Gao *et al.* BRCA-paclitaxel data set (see "Threshold Section on BRCA-paclitaxel data" section), threshold-selected to contain tumor models that behave in a completely responsive manner (Fig 2, Steps 6, 7a, and 7b). In this data set, the SMSL screening test has a sensitivity of 0.96 at a significance level of 0.1 (24/25 p-values less than 0.10; clear cells), and 0.92 at 0.05 significance level (23/25 p-values less than 0.05; clear cells). Those p-values in bold are ones that are greater than 0.001. However, every novel screening test's sensitivity is 1.00 over all significance levels, for the same reasons as those observed in Table 7.

As with the results in Fig 4 for the Evrard *et al.* data, the 95% CIs for Fig 5 show different lengths for the different drug-screening tests and significance levels. Furthermore, there are clearly superior drug screening methods in terms of median sensitivities and 95% CI lengths.. The screening methods are NMNL and NMSL. The median sensitivities and 95% CIs are 1.0 (interval lengths = 0.0) for all significance levels. In fact, the 99% CIs are 1.0 for all significance levels as well.

**Table 8. The p- and q-values generated from the Gao *et al.* Completely Responsive (CR) data set.**

| | | Lab | | | | | |
|---|---|---|---|---|---|---|---|
| | Measure | Simulated Lab 1 | Simulated Lab 2 | Simulated Lab 3 | Simulated Lab 4 | Simulated Lab 5 | Fisher's Method |
| p-values | $DTV_{21}$ | **3.50E-03** | **3.95E-03** | 3.42E-05 | 3.57E-04 | **1.63E-03** | 2.12E-11 |
| | $AUC_{21}$ | 3.52E-06 | 1.85E-06 | 4.32E-10 | 1.08E-07 | 3.35E-07 | 1.68E-28 |
| | AUCmax | 9.97E-05 | 1.46E-04 | 1.81E-07 | 1.43E-05 | 1.41E-04 | 1.97E-18 |
| | $TGI_{21}$ | 2.11E-04 | 4.04E-05 | 1.81E-07 | 1.43E-05 | 1.41E-04 | 1.20E-18 |
| | $PFS_4$ | **1.48E-01** | **3.17E-02** | **2.20E-03** | **3.20E-02** | **7.01E-02** | 1.17E-04 |
| q-values | $DTV_{21}$ | 3.50E-03 | 4.93E-03 | 3.42E-05 | 3.57E-04 | 1.63E-03 | 2.64E-11 |
| | $AUC_{21}$ | 3.52E-06 | 9.27E-06 | 4.32E-10 | 1.08E-07 | 3.35E-07 | 8.40E-28 |
| | AUCmax | 9.97E-05 | 2.43E-04 | 1.81E-07 | 1.43E-05 | 1.41E-04 | 3.28E-18 |
| | $TGI_{21}$ | 2.11E-04 | 1.01E-04 | 1.81E-07 | 1.43E-05 | 1.41E-04 | 3.00E-18 |
| | $PFS_4$ | 1.48E-01 | 3.17E-02 | 2.20E-03 | 3.20E-02 | 7.01E-02 | 1.17E-04 |

The values in the white cells are the results of the 25 SMSL tests carried out. There are additionally 5 SMNL test results (light gray), 5 NMSL test results (dark gray), and 1 NMNL test result (steel gray) in this table. The values in bold are SMSL test results where the p-value (25 clear cells) is greater than 0.001.

%ΔVol(t): The percent relative change from day t = 0 of the tumor growth at day t.

$DTV_t$: %ΔVol(t).

Simulated Labs 1–5: Independent and random Bootstrap Resamples of the Gao et al dataset.

SMSL: Single-Measure, Single-Lab.

SMNL: Single-Measure, Numerous-Lab.

NMSL: Numerous-Measures, Single-Lab.

NMNL: Numerous-Measures, Numerous-Labs.

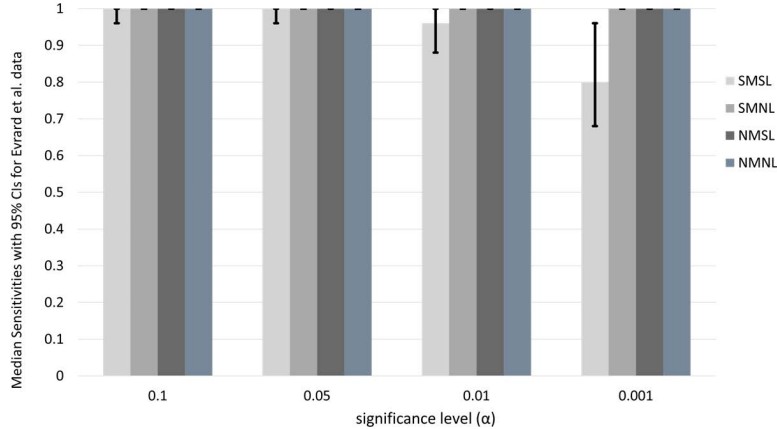

**Fig 4. Median sensitivity of drug screening tests at varying significance levels for Evrard *et al.* bootstrap data.** Drug screening-test median sensitivities and 95% CIs for Evrard *et al.* data. All values are determined from 10,000 bootstrap samples of the original dataset. Confidence intervals are indicated by vertical lines for each test and significance level. The height of each bar is the median sensitivity over all bootstrap samples.

The SMSL drug-screening test results for the Gao *et al.* data are similar to those for the Evrard *et al.* data. The SMSL test has the lowest median sensitivity and is the least precise of all four drug-screening tests (all 95% CI-lengths greater than 0). We glean this information from Fig 5. As with Fig 4, the median sensitivity for the SMSL decreases as α decreases. With one exception, the SMSL 95% CI length is greater than that of all other screening tests. The exception is for the SMNL screening test at α = 0.001, where both tests have a 95% CI length of 0.20.

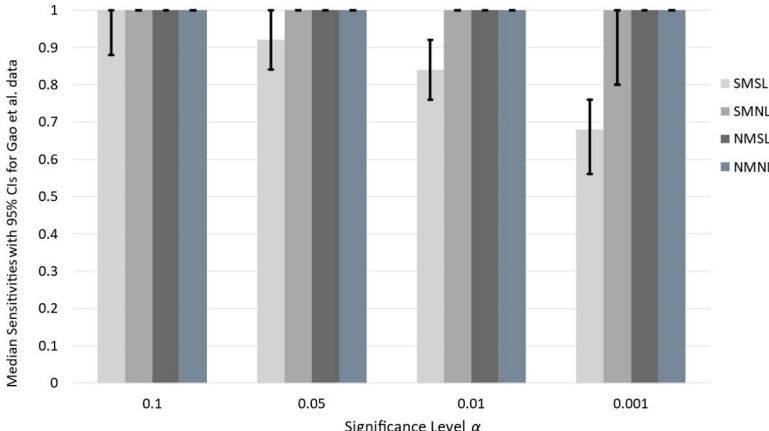

**Fig 5. Median sensitivity of drug screening tests at varying significance levels for Gao _et al._ bootstrap data.** All values are determined from 10,000 bootstrap samples of the original dataset. Confidence intervals are indicated by vertical lines for each test and significance level. The height of each bar is the median sensitivity over all bootstrap samples.

From all this information, we draw the conclusion that, with 95% confidence, the NMNL and NMSL sensitivities are 1.00 for all significance levels for this data set with a CI length of zero. By comparing these results with those in Fig 3, we conclude that these two tests are optimal in terms of sensitivity and precision for the Evrard _et al._ and Gao _et al._ data.

A second choice is the SMNL screening test. For the Evrard _et al._ data set, results are the same as for the NMNL and NMSL tests, in terms of sensitivity and precision. For the Gao _et al._ data, sensitivity and precision are like those for the other two tests for $\alpha = 0.1$, 0.05, and 0.01. For significance level 0.001 the median sensitivity is 1.00, and as noted above, the 95% CI is [0.8, 1]. While not as powerful or precise as the NMNL or NMSL tests, we are 95% confident that the SMNL sensitivity is at least 0.80 for $\alpha = 0.001$.

## Specificity

**Evrard _et al._ data basis tables with novel drug screening test results for progressive disease data sets.** When determining the specificity of drug-screening tests, we ask how often different laboratories, using different sets of measure-related statistical tests, classify a tumor as progressive disease (PD) if it has a ground-truth classification of being PD.

We present the specificity results for the PD data set from the Evrard _et al._ publication in Table 9 (Fig 2, Steps 6, 7a, and 7b).

We provide the clustered bar chart of median specificities for all four drug screening tests at a variety of significance levels in Fig 6.

Given the information from Table 9 and Fig 6, we determine that the SMNL, NMSL, and NMNL test median specificities are 1.0 for all significance levels. After rounding to two digits, the median SMSL specificities are: 0.82 (18/22), 0.95 (21/22), 1.0 (22/22), 1.0 (22/22).

We might conclude from this result that the best performing drug-screening test for specificity is the SMSL test. The upper confidence limit (UCL) for the SMSL test matches that of all other drug screening tests. Additionally, it is more precise (95% CI length is shorter) than all other screening tests at every significance level, except for the NMSL test at 0.05, whose CI length is slightly less than the SMSL.

It is expected that the specificity increases as α decreases. The p-values are fixed in any table like Table 9, including any bootstrap sampled table. As we make the significance level α more stringent, the proportion of p and q-values that are greater than α will either increase or remain the same, as observed previously.

**Table 9. P-values and q-values for the Evrard *et al.* Progressive Disease (PD) data set.**

| | | Lab | | | | | |
|---|---|---|---|---|---|---|---|
| | **Measure** | **HCI-BCM** | **MDACC** | **PDMR** | **WUSTL** | **WIST** | **Fisher's Method** |
| p-values | $DTV_{21}$ | 1.43E-01 | 1.63E-01 | 4.04E-01 | 9.18E-01 | NA | 3.02E-01 |
| | $AUC_{21}$ | **7.13E-02** | 2.49E-01 | 6.85E-01 | 3.50E-01 | NA | 2.06E-01 |
| | AUCmax | 1.77E-01 | 3.05E-01 | 5.21E-01 | 4.69E-01 | 3.60E-01 | 3.81E-01 |
| | $TGI_{21}$ | **6.75E-02** | **6.65E-02** | 7.51E-01 | 5.38E-01 | NA | 1.25E-01 |
| | $PFS_4$ | 4.17E-01 | **2.84E-02** | 7.78E-01 | 2.00E-01 | 4.80E-01 | 1.70E-01 |
| q-values | $DTV_{21}$ | 2.21E-01 | 2.72E-01 | 7.78E-01 | 9.18E-01 | NA | 3.77E-01 |
| | $AUC_{21}$ | 1.78E-01 | 3.05E-01 | 7.78E-01 | 6.73E-01 | NA | 3.44E-01 |
| | AUCmax | 2.21E-01 | 3.05E-01 | 7.78E-01 | 6.73E-01 | 4.80E-01 | 3.81E-01 |
| | $TGI_{21}$ | 1.78E-01 | 1.66E-01 | 7.78E-01 | 6.73E-01 | NA | 3.44E-01 |
| | $PFS_4$ | 4.17E-01 | 1.42E-01 | 7.78E-01 | 6.73E-01 | 4.80E-01 | 3.44E-01 |

The four colored portions of the table, corresponding to the four drug screening tests, are the same as that in Table 3. Those p-values in bold are less than 0.10. Note that there are no p-values for the Wistar lab with measures that require values at t = 21 days. For Wistar, measurements for each mouse were stopped at t = 14 days due to the design protocol of the Evrard *et al.* study at that location. Cells containing the text "NA" indicate that the p-value could not be computed for the specific measure and lab.

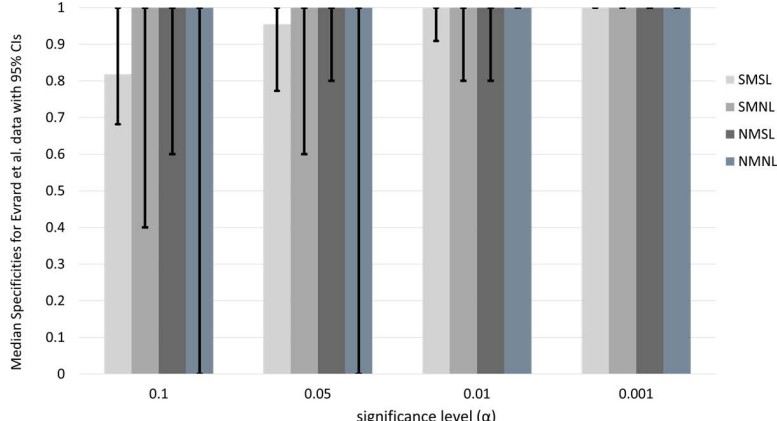

**Fig 6. Median specificity of drug screening tests at varying significance levels for the Evrard *et al.* data.** Median specificities are indicated by the heights of the bars. For each significance level, there are four bars above; the light gray, gray, dark gray, and steel gray bars, denoting respective specificities for the SMSL, SMNL, NMSL, and NMNL drug screening tests. The vertical line segments in the middle of each bar provide the endpoints of the bootstrap 95% CIs.

Because we observed strong results for the NMNL drug screening test under sensitivity for the Evrard *et al.* data (see Fig 4), its behavior under specificity is unusual. The 95% bootstrap CIs are [0,1] for the $\alpha = 0.10$ and 0.05 significance levels for the NMNL (Fig 6). This offers a potentially misleading conclusion that the NMNL screening test is non-specific. However, the NMNL test simply cannot take on any specificity value other than 0 or 1, the extremes of the interval. Further examination of the results showed that the rate of false positives in the NMNL test are commensurate with the type I error rate corresponding to the significance level (proportion of false positives ≈ α). Combining the results of Figs 5 and 6, we have a strong argument for setting the type I error rate to 0.01 for the novel drug screening tests for the Evrard *et al.* data, and using the NMNL screening test.

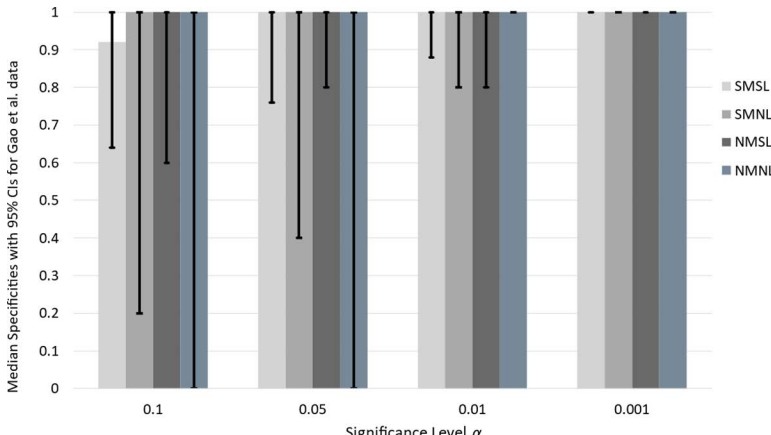

**Fig 7. Median specificity of drug screening tests at varying significance levels for the Gao *et al.* data.** Median specificities are indicated by the heights of the bars. For each significance level, there are four bars above; the light gray, gray, dark gray, and steel gray bars, denoting respective specificities for the SMSL, SMNL, NMSL, and NMNL drug screening tests. The vertical line segments in the middle of each bar provide the endpoints of the bootstrap 95% CIs.

**Gao *et al.* data basis tables with novel drug screening test results for progressive disease data sets.** Due to the virtually identical results for the Gao *et al.* specificity basis table and that for the Evrard *et al* table (Table 9), we omit it from this section.

Clustered bar charts and 95% CIs determined from the 10,000 Gao *et al.* specificity bootstrap samples are presented in Fig 7. The findings are similar to those in Fig 6 for the Evrard *et al.* data. All drug-screening tests have positive 95% CIs for all significance levels for significance levels 0.10 and 0.05. The NMNL drug-screening test has a 95% CI of 1.0 for significance levels 0.01 and 0.001. In fact, all drug-screening tests have a 95% CI of 1.0 at α = 0.001. This result is the same as for the respective drug-screening tests and significance level with the Evrard *et al.* data (Fig 6). We can make the same point about NMNL being the optimal statistic at α = 0.01, and one of the optimal screening tests for α = 0.001 with the Gao *et al.* data.

As noted above, in S1 File, we perform all of the steps in Fig 2 for an actual data set. Given the strong performance of the meta-analysis tests SMNL and NMNL, we provide a summary of median sensitivities and specificities for those tests in Table 10.

A result that comes from studying Table 10 is that, for sensitivity, SMNL and NMNL screening tests are superior. For every significance level and both datasets, the median sensitivity is 1.0, and the 95% confidence interval is (1.0,1.0), meaning 95% of the bootstrap samples have a sensitivity of 1.0. For specificity, optimal results occurred for the 0.01 and 0.001 significance levels for the NMNL (both datasets) and for 0.001 for the SMNL (apart from the 0.001 significance level, Gao *et al.* dataset). It follows that, to obtain the highest sensitivity and specificity jointly, we select the NMNL screening test and specify a significance level of 0.01.

**Accuracy of drug-screening tests for heterogeneous laboratory data.** When considering sensitivity and specificity, the ground truth classification and the corresponding p-values and q-values for each lab in the Evrard *et al.* data is the same (Tables 7 and 9 above). The reason is that each lab was given the same tumor models and drug treatments. It is critical to consider more general situations in which the ground truth classifications are not known or are a mixture for different labs. Such situations include heterogeneity of tumor model samples across multiple laboratories, different labs having different protocols, or an effect size sufficiently small that p-values will not be informative. For sensitivity and specificity, in this case we are not performing an apples-to-apples comparison.

**Table 10. Median sensitivities, specificities, and 95% confidence intervals for the SMNL and NMNL drug-screening tests for the Evrard *et al.* and Gao *et al.* PDX datasets.**

| | | Evrard *et al.* | | | | Gao *et al.* | | | |
|---|---|---|---|---|---|---|---|---|---|
| **Significance Level** | | 0.1 | 0.05 | 0.01 | 0.001 | 0.1 | 0.05 | 0.01 | 0.001 |
| Sensitivity | SMNL | **1.0 (1.0,1.0)** | **1.0 (1.0,1.0)** | **1.0 (1.0,1.0)** | **1.0 (1.0,1.0)** | **1.0 (1.0,1.0)** | **1.0 (1.0,1.0)** | **1.0 (1.0,1.0)** | 1.0 (0.8,1.0) |
| | NMNL | **1.0 (1.0,1.0)** | **1.0 (1.0,1.0)** | **1.0 (1.0,1.0)** | **1.0 (1.0,1.0)** | **1.0 (1.0,1.0)** | **1.0 (1.0,1.0)** | **1.0 (1.0,1.0)** | **1.0 (1.0,1.0)** |
| Specificity | SMNL | 1.0 (0.4,1.0) | 1.0 (0.6,1.0) | 1.0 (0.8,1.0) | **1.0 (1.0,1.0)** | 1.0 (0.2,1.0) | 1.0 (0.4,1.0) | 1.0 (0.8,1.0) | **1.0 (1.0,1.0)** |
| | NMNL | 1.0 (0.0,1.0) | 1.0 (0.0,1.0) | 1.0 (1.0,1.0) | **1.0 (1.0,1.0)** | 1.0 (0.0,1.0) | 1.0 (0.0,1.0) | **1.0 (1.0,1.0)** | **1.0 (1.0,1.0)** |

SMNL: Single-Measure, Numerous-Labs. NMNL: Numerous-Measures, Numerous-Labs. For each cell, the median sensitivity or specificity is outside the parentheses, and the two values inside are the lower and upper 95% confidence interval values, respectively. Those cells in bolds have median sensitivity or specificity of 1.0, and their 95% CI is a single value, 1.0.

In such situations, one statistic of interest is the accuracy of the drug screening test. Using notation from Table 5, accuracy is defined as:

$$\frac{TP + TN}{P + N}.$$

Therefore, accuracy is useful in this situation because it does not require homogeneous ground-truth classifications across labs, unlike sensitivity and specificity. If there is only a completely responsive tumor type, then N and TN are 0, and accuracy reduces to sensitivity. Conversely, if there is only a progressive disease tumor type, then P and TP are 0, and accuracy reduces to specificity.

As an example of how our drug screening tests perform when either the ground truth is not known or is a mixture of CR and PD classifications for different labs, we constructed a mixed ground-truth data set from the Evrard *et al.* data. The actual conditions selected for the data set are listed in Table 11 below. The basis table corresponding to Table 11 was constructed by using the BCM laboratory p-values and q-values from Table 9 (PD), the MDA laboratory p-values and q-values from Table 7 (CR), and so forth.

To demonstrate our findings at multiple significance levels, we include histograms of the accuracy for the four drug screening tests across 10,000 bootstrap resamples. In Figs 8–11, we provide histograms for significance levels 0.1, 0.05, 0.01, and 0.001. The range of accuracy considered is 0.7 to 1.00, since there was no accuracy less than 0.70 for any screening test at any significance level.

From our study of these figures, there are several key findings. The first is that the accuracy of the SMNL and NMNL tests are always 1.00 for all significance levels (all bootstrap samples have an accuracy of 1.0). For these two screening tests, the accuracy reduces to sensitivity, since three of the labs in Table 11 have ground truths that are completely responsive.

Another finding is that the proportion of NMSL results with 1.00 accuracy is less than 100% for significance level 0.1 (Fige 8) and slowly increases to 100% as the significance level decreases (Figs 9–11). It appears that if the NMSL accuracy is not 1.00, it is 0.85 (Figs 8 and 9).

**Table 11. Mixed ground truth classification data set used to compute accuracy.**

| Lab | BCM | MDA | PDMR | WUSTL | WIST |
|---|---|---|---|---|---|
| **Constructed ground-truth classification** | PD | CR | CR | PD | CR |

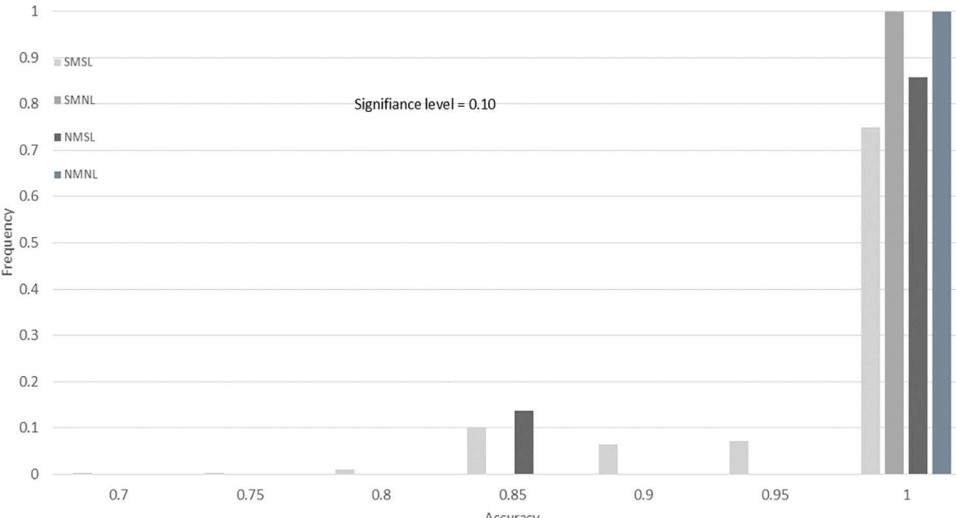

**Fig 8. Drug screening test accuracy for significance level 0.10.** This histogram is created using 10,000 bootstrap resamples of the data from Evrard *et al.*, with lab-specific ground truth classifications in Table 11.

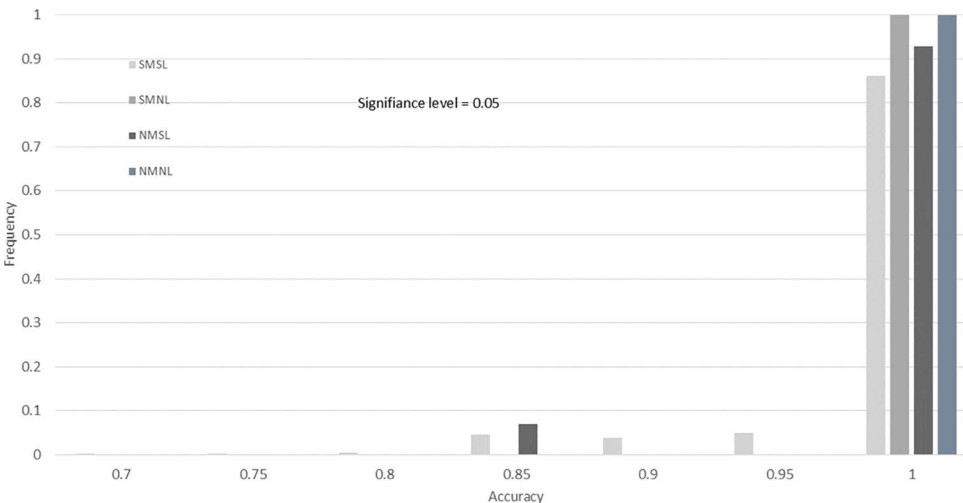

**Fig 9. Drug screening test accuracy for significance level 0.05.** This histogram is created using 10,000 bootstrap resamples of the data from Evrard *et al.*, with lab-specific ground truth classifications in Table 11.

Finally, the SMSL test appears to have no clear convergence properties. The proportion of bootstraps for which the SMSL test has an accuracy of 1.0 increases from 0.75 for about 0.9 for significance levels 0.10 to 0.01 (Figs 8–10). Then the proportion drops to 0.18 for $\alpha = 0.001$. The highest proportion of accurateness for the SMSL test at the 0.001 significance level is 0.85, with 45% of the bootstrap samples having that accuracy.

Our results for the heterogeneous data point to more general applications of our drug-screening tests, where the laboratory-specific assignments of CR and PD ground-truth classifications will differ from those in Table 11.

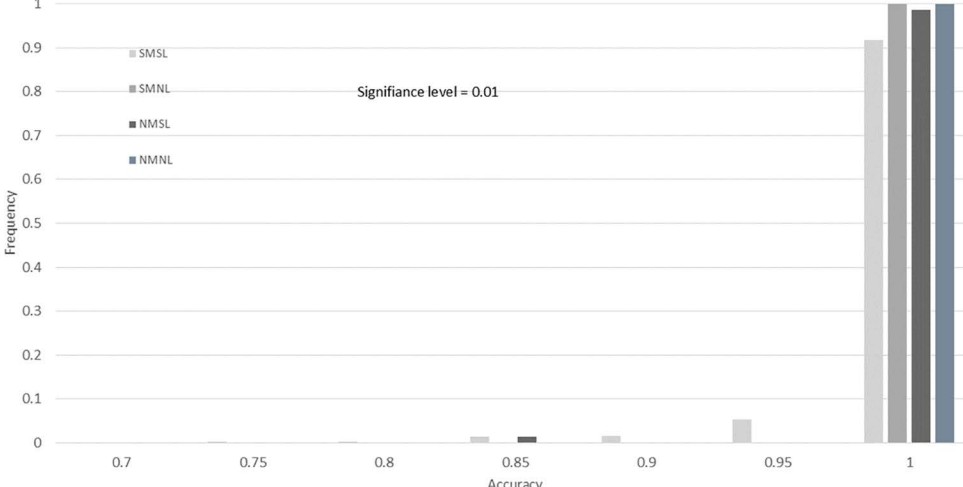

**Fig 10. Drug screening test accuracy for significance level 0.01.** This histogram is created using 10,000 bootstrap resamples of the data from Evrard *et al.*, with lab-specific ground truth classifications in Table 11.

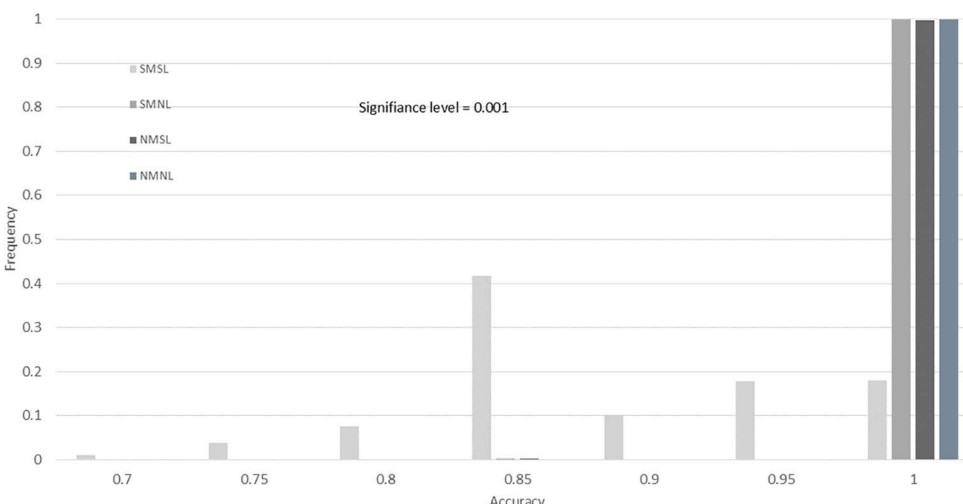

**Fig 11. Drug screening test accuracy for significance level 0.001.** This histogram is created using 10,000 bootstrap resamples of the data from Evrard *et al.*, with lab-specific ground truth classifications in Table 11.

## Discussion

We have shown in this paper that drug screening tests that incorporate numerous tumor growth measures and meta-analysis across numerous independent experiments are equivalent to or an improvement on the Single-Measure, Single-Lab screening test in terms of sensitivity, specificity, and accuracy. We therefore recommend using multiple tumor growth measures and multiple experiments when evaluating candidate cancer drug effectiveness in PDX trials.

We have particularly shown the advantage of using numerous measures of drug inhibition of tumor growth in patient-derived xenografts. The advantage of using numerous measures of drug inhibition of tumor growth is not limited to patient-derived xenografts. The method described here can be applied to other situations in which drug inhibition needs

to be reliably evaluated. These situations include tumors derived from cell lines [36] and organoids [37], as well as tumors that arise in genetically engineered mice [38,39] and carcinogen-treated mice [40]. The method could also improve the reliability of determining drug inhibition of cells grown in tissue culture in two dimensions or three dimensions [41–43].

Some published findings in preclinical cancer biology cannot be replicated, either due to incomplete records of experimental protocols, small effect sizes, or large false positive rates, e.g., from incomplete correction for multiple testing [44]. Addressing the replicability problem requires, among other solutions, the development of simple, intuitive statistical methods for analysis that can maintain robustness in the presence of heterogeneity due to variation in experimental protocol. It is possible to ensure high statistical power and low false positive rates by using several statistical growth measures applied to data obtained from more than one laboratory. Our method of using numerous measures of tumor growth includes individual measures that together can also account for non-monotonic tumor growth trajectories, missing time observations, and censored data. For instance, the Area Under the Curve (AUC) is non-parametric, the Tumor Growth Index ($TGI_{21}$) needs observations at only 0 days and 21 days so observations at intermediate times may be missing, and Progression-Free Survival (PFS) may be right-censored. The numerous measures method uses a combination of conventional statistical tests that are familiar to most experimentalists.

It is possible to conduct multiple experiments in a single study in an empirical manner. The simplest approach is to divide the treatment and control tumor populations equally into two cohorts, and to test the candidate cancer drug on the separate treatment cohorts on different days. If the staffing is available, even greater independence can be guaranteed by having one team test one cohort, and a second team test the other. By conducting at least two trials on separate cohorts, one can take advantage of Fisher's method or another p-value combining method to increase drug screening test robustness.

Even if it is not feasible to conduct multiple independent experiments, we additionally showed that even the Numerous-Measures, Single-Lab screening test has equal or greater sensitivity and specificity to the Single-Measure, Single-Lab screening test. Calculating tumor growth measures based on the tumor volume over time incurs no additional cost in time and materials over the standard analysis plan. One caveat is that, for ideal rigor, numerous measures of tumor growth should not be presented without multiple testing correction. An additional caveat for the NMSL test is that there are no replicated data, so it is not possible to infer results across data sets; for this reason, we recommend doing at least one additional experiment per study, for example, repeating the same study at a different time.

Meta-analysis of multiple measures of tumor inhibition by anti-cancer drugs can have broader application than to the PDX models described here. Recently, several new *in vitro* models have been described and proposed as platforms for screening new candidate drugs or as patient avatars for precision medicine. These include tumor organoids [37], patient-derived explants [45], patient-derived micro-organospheres [46], and organs-on-a-chip [47]. Each of these models have specific advantages and limitations [6,7]. However, compared to *in vivo* PDXs, the *in vitro* models promise to be less expensive, provide more rapid results, and are amenable to high-throughput assays. For each of these *in vitro* platforms, meta-analysis of multiple measures of tumor inhibition by anti-cancer drugs validated in different laboratories could indicate which measures of inhibition are most robust with greatest sensitive and specificity.

In addition to the classical statistical tests for tumor growth, other tests have been devised. Gao *et al.* [5] described a BestAvgResponse of tumor volume (V) to drug treatment. BestResponse is the minimum value of %ΔVol(t) for t ≥ 10 days. The BestAvgResponse is an average metric that indicates a combination of speed, strength, and durability of tumor growth and drug inhibition.

Heitjan [48] provided a useful review comparing several classical statistical tests applied to tumor growth data in animals, including the assumptions for and an informative critique of each test. Leffondré *et al.* [49] developed statistical measures for longitudinal data that discriminate between stable-unstable, increasing-decreasing, linear-nonlinear, and monotonic-nonmonotonic trajectories. This allows distinguishing groups of trajectories that regularly decrease, are stable, highly unstable, or have abrupt changes. Tan *et al.* [50] account for incomplete and missing tumor growth data using a

maximum likelihood method based upon the expectation/conditional maximization (ECM) algorithm. Liang [51] applied a nonparametric linear mixed-effects model to estimate the curves of tumor volumes with time. Wu and Houghton [52] suggested using a nonparametric bootstrap percentile interval of the Log10 cell kill (LCK) rather than an arbitrary cutoff of the LCK, and assessing the effect of cytotoxic treatment by the confidence limits of the LCK. Roy Choudhury et al. [53] account for a change in the effect of therapy over time by using a piecewise quadratic model with flexible boundaries. Demidenko et al. [54] described exponential growth and regrowth using three endpoints, doubling time (DT), tumor growth delay (TGD), and cancer surviving fraction. Medioni et al. [55] proposed two new parameters, Time to Relapse (TTR) and Tumor Growth Speed (TGS), to overcome the limitations of the conventional Tumor Growth Index (TGI) and Tumor Growth Delay Index (TGDi). Pan et al. [56] proposed joint modeling of longitudinal tumor growth curve data and survival data, using a Markov chain Monte Carlo approach to estimate the parameters of the joint model. Laajala et al. [57] were able to detect subtle treatment effects even in the presence of high within-group variability by using an expectation maximization algorithm coupled with a mixed-effects modeling framework. Hather et al. [58] extend the traditional T/C ratio, which uses single time measurements, to a rate-based T/C ratio which uses measurements of the exponentially transformed data at all times. Corwin et al. [59] proposed a new composite parameter, Tumor Control Index (TCI) composed of the Tumor Inhibition Score, Tumor Rejection Score, and Tumor Stability Score. Zhao et al. [60] developed a Bayesian hierarchical change point method that accounts for non-monotonic tumor profiles by describing prenadir, postnadir, nadir and regression periods.

The threshold-selection technique is useful for establishing a proxy for ground truth where the outcome of the study is not already known. Often, threshold selection is used to classify the top 50% and bottom 50% of the population when the sample size is 5 or fewer in each cohort. When the total population is 10 or more in each cohort, or 20 tumor models total, selecting smaller sub-populations such as the top 25% and bottom 25% provides a good approximation for complete response and progressive disease classification among a combined cohort of treatment and controls together. This serves a dual purpose: it excludes control tumors that grew poorly during the study and weeds out treated tumors that did not respond similarly to the candidate cancer drug as their replicates. Further nuance may be required when sub-sampling across tumor models from more than one PDX sample, as failure to grow or failure to respond may have meaning.

Future directions include evaluating drug screening test performance at scale across various cancer types. This will confirm our initial findings on robustness for the drug screening tests. By robustness, we mean considering other parameter settings than those we have considered in this paper, such as differing number of labs, number of measures, type of cancer, or the number of mice in the study. We can also repeat the analysis for different data sets or cancer types. We can also extend this type of drug screening test to other types of candidate cancer drug experiments, such as those previously cited, or in other types of studies such as tumor organoids, organ-on-a-chip, or alternative animal models such as *C. elegans* or zebrafish.

## Conclusion

Our initial meta-analysis results—accuracy, sensitivity, and specificity—with real-world data indicate that the most powerful and reliable procedure to characterize a drug's ability to inhibit tumor growth in a PDX trial is to use several different measures of tumor growth collected at multiple different laboratory sites. If data are collected at only one laboratory site, then multiple independent experiments should be done on different days and analyzed with several different measures of tumor growth.

## Supporting information

**S1 File. Example PDX statistical calculations.** A step-by-step illustration of how the four screening tests are scored for an example data set.
(DOCX)

**S2 Table. Empirical Tumor Growth Trajectories for Evrard *et al.* Completely Responsive Tumor Models.** The data points (time, Percent relative change in tumor growth) for each mouse in each treatment group for a completely responsive tumor model.
(XLSX)

**S3 Table. Empirical Tumor Growth Trajectories for Evrard *et al*. Progressive Disease Tumor Models.** The data points (time, Percent relative change in tumor growth) for each mouse in each treatment group for a progressive disease tumor model.
(XLSX)

**S4 Table. Empirical Tumor Growth Trajectories for Gao *et al.* Progressive Disease Tumor Models.** The data points (time, Percent relative change in tumor growth) for each mouse in each treatment group for a progressive disease tumor model.
(XLSX)

**S5 Table. Empirical Tumor Growth Trajectories for Evrard *et al.* Completely Responsive Tumor Models.** The data points (time, Percent relative change in tumor growth) for each mouse in each treatment group for a completely responsive tumor model.
(XLSX)

**S6 Table. Bootstrap samples for all Evrard *et al*. and Gao *et al*. Drug Screening Tests.** The collection of all bootstrap samples for all drug screening tests with empirical tumor growth trajectories for Evrard *et al*. and Gao *et al*., Completely Responsive and Progressive Disease tumor models.
(XLSX)

## Acknowledgments

The authors thank Michael Lloyd for furnishing patient-derived xenograft data used in the manuscript "Systematic Establishment of Robustness and Standards in Patient-Derived Xenograft Experiments and Analysis" authored in 2020 by Evrard *et al.* We additionally acknowledge the authors of "High-throughput screening using patient-derived tumor xenografts to predict clinical trial drug response" authored in 2015 by Gao *et al.*, for providing their patient-derived xenograft data freely as a supplement to their publication.

## Author contributions

**Conceptualization:** Elizabeth Rosenzweig, David E. Axelrod, Derek Gordon.

**Data curation:** Elizabeth Rosenzweig.

**Formal analysis:** Elizabeth Rosenzweig, David E. Axelrod.

**Investigation:** David E. Axelrod, Derek Gordon.

**Methodology:** Elizabeth Rosenzweig, David E. Axelrod, Derek Gordon.

**Project administration:** David E. Axelrod.

**Resources:** David E. Axelrod.

**Software:** Elizabeth Rosenzweig.

**Supervision:** David E. Axelrod.

**Validation:** Elizabeth Rosenzweig.

**Writing – original draft:** Elizabeth Rosenzweig, David E. Axelrod, Derek Gordon.

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
