## [Decision Letter · Decision Letter 0]

7 Mar 2025

PONE-D-24-51054Improved drug-screening tests of candidate anti-cancer drugs in patient-derived xenografts through use of numerous measures of tumor growth determined in multiple independent laboratoriesPLOS ONE

Dear Dr. Gordon,

Thank you for submitting your manuscript to PLOS ONE. After careful consideration, we feel that it has merit but does not fully meet PLOS ONE’s publication criteria as it currently stands. Therefore, we invite you to submit a revised version of the manuscript that addresses the points raised during the review process.

We look forward to receiving your revised manuscript.

Kind regards,

Afzal Basha Shaik, Ph.D

Academic Editor

PLOS ONE

Reviewers' comments:

Reviewer's Responses to Questions

**Comments to the Author**

1. Is the manuscript technically sound, and do the data support the conclusions?

Reviewer #1: Yes

Reviewer #2: Yes

2. Has the statistical analysis been performed appropriately and rigorously? 

Reviewer #1: Yes

Reviewer #2: Yes

3. Have the authors made all data underlying the findings in their manuscript fully available?

Reviewer #1: Yes

Reviewer #2: Yes

4. Is the manuscript presented in an intelligible fashion and written in standard English?

Reviewer #1: Yes

Reviewer #2: Yes

5. Review Comments to the Author

Reviewer #1: Elizabeth et al. reported Improved drug-screening tests of candidate anti-cancer drugs in patient-derived xenografts through use of numerous measures of tumor growth determined in multiple independent laboratories. The goal of the work was to document improvements in the experimental design and statistical analysis of drug-screening tests based upon the criteria of sensitivity, specificity, and accuracy. They generated a drug-screening test based on a single measure and a single lab (SMSL; single p-value) and (Fisher’s Method p-values; one p-value per measure) and/or Storey’s method (m q-values for m measures in a single lab). They revealed that, for these data, the optimal drug screening tests employ the meta-analysis approach, while the SMSL produces the greatest uncertainty. meta-analysis results (accuracy, sensitivity, and specificity) indicate that the most reliable procedure to characterize the ability of a chemotherapeutic drug to inhibit tumor growth in a patient-derived xenograft is to use several different measures of tumor growth collected at multiple different laboratory sites. However, if data are collected at only one laboratory site, then multiple independent experiments should be done on different days and analyzed with several different measures of tumor growth

The manuscript is scientifically sound considering the need to develop newer drug screening tests in patient-derived xenografts. Its well drafted and the statistical analysis have been performed effectively. However, there are certain modifications to be made to increase the quality and readership of the article. Firstly, the authors should include the abbreviations below all the tables. For example, in Tables 7, 9 and 10, the abbreviations of the corresponding labs are suggested to be incorporated in the revised manuscript. Secondly, the authors can also mention if there were any similarities in both the datasets obtained from different labs. Thirdly, it would be ideal to summarize the sensitivity and specificities of the screening methods of meta analysis in a single table under results/conclusion section. Lastly, the figures look somewhat ‘blurry’ and quality of the figures can be enhanced.

Reviewer #2: The present work entitled: Improved drug-screening tests of candidate anti-cancer drugs in patient-derived xenografts through the use of numerous measures of tumor growth determined in multiple independent laboratories that may be useful as prospective anticancer agents. The purpose of this work is to develop methods for evaluating the results of preclinical drug screening tests. Several different measures of drug inhibition of tumor growth in patient-derived xenografts (PDX) are compared, and the benefit of determining inhibition in different laboratories are compared. The work is interesting and suitable for publication in the PLOS ONE after making minor revisions. I suggest the authors to make the necessary corrections as suggested below

1. Rewrite the keywords in alphabetical order. Additionally, add the other keywords preclinical, in vivo and invitro

2. Check the manuscript for grammatical and typographical errors. For example, keep the word et al, in vivo, in vitro in italics, and while writing the word figure “F” must be capital letter.

3. Table legends of all the tables mentioned in the manuscript have to be keep above the table

4. Double check the entire documents, different typos are present

5. To improve the quality of manuscript, add some of the recent findings related to anti-cancer in introduction and results

Huang, H., Huang, F., Liang, X., Fu, Y., Cheng, Z., Huang, Y.,... Chen, Y. (2023). Afatinib Reverses EMT via Inhibiting CD44-Stat3 Axis to Promote Radiosensitivity in Nasopharyngeal Carcinoma. Pharmaceuticals, 16(1), 37. doi: https://doi.org/10.3390/ph16010037

Wan, H., Zhou, S., Li, C., Zhou, H., Wan, H., Yang, J.,... Yu, L. (2024). Ant colony algorithm-enabled back propagation neural network and response surface methodology based ultrasonic optimization of safflower seed alkaloid extraction and antioxidant. Industrial Crops and Products, 220, 119191. doi:https://doi.org/10.1016/j.indcrop.2024.119191

Yao, X., Zhu, Y., Huang, Z., Wang, Y., Cong, S., Wan, L.,... Hu, Z. (2024). Fusion of shallow and deep features from 18F-FDG PET/CT for predicting EGFR-sensitizing mutations in non-small cell lung cancer.

Quantitative Imaging in Medicine and Surgery 2024, 14(8), 5460-5472. doi: 10.21037/qims-23-1028Hu, Y., Zhang, Q., Bai, X., Men, L., Ma, J., Li, D.,... Xie, T. (2024). Screening and modification of (+)-germacrene A synthase for the production of the anti-tumor drug (−)-β-elemene in engineered Saccharomyces cerevisiae. International Journal of Biological Macromolecules, 279, 135455. doi:https://doi.org/10.1016/j.ijbiomac.2024.135455

Zhu, J., Jiang, X., Luo, X., Zhao, R., Li, J., Cai, H.,... Xie, T. (2023). Combination of chemotherapy and gaseous signaling molecular therapy: Novel β-elemene nitric oxide donor derivatives against leukemia.

Drug Development Research, 84(4), 718-735. doi: https://doi.org/10.1002/ddr.22051

Li, J., Chen, Y., Zhang, S., Zhao, Y., Gao, D., Xing, J.,.Xu, G. (2025). Purslane (Portulaca oleracea L.) polysaccharide attenuates carbon tetrachloride-induced acute liver injury by modulating the gut microbiota in mice. Genomics, 117(1), 110983. doi: https://doi.org/10.1016/j.ygeno.2024.110983

Lodi, R. S., Dong, X., Wang, X., Han, Y., Liang, X., Peng, C.,... Peng, L. (2025). Current research on the medical importance of Trametes species. Fungal Biology Reviews, 51, 100413. doi: https://doi.org/10.1016/j.fbr.2025.100413

Zhang, D., Song, J., Jing, Z., Qin, H., Wu, Y., Zhou, J.,... Zang, X. (2024). Stimulus Responsive Nanocarrier for Enhanced Antitumor Responses Against Hepatocellular Carcinoma. International Journal of Nanomedicine, 19, 13339-13355. doi: https://doi.org/10.2147/IJN.S486465

Du, F., Ye, Z., He, A., Yuan, J., Su, M., Jia, Q.,... Wang, Z. (2025). An engineered α1β1 integrin-mediated FcγRI signaling component to control enhanced CAR macrophage activation and phagocytosis. Journal of Controlled Release, 377, 689-703. doi: https://doi.org/10.1016/j.jconrel.2024.11.064

Dong, Q., & Jiang, Z. (2024). Platinum–Iron Nanoparticles for Oxygen-Enhanced Sonodynamic Tumor Cell Suppression. Inorganics, 12(12), 331. doi: https://doi.org/10.3390/inorganics12120331

Zhao, C., Song, W., Wang, J., Tang, X., & Jiang, Z. (2025). Immunoadjuvant-functionalized metal–organic frameworks: synthesis and applications in tumor immune modulation. Chemical Communications, 61(10), 1962-1977. doi: 10.1039/D4CC06510G

Zeng, Q., Jiang, T., & Wang, J. (2024). Role of LMO7 in cancer (Review). Oncol Rep, 52(3), 117. doi:10.3892/or.2024.8776

Wu, N., Zhang, X., Fang, C., Zhu, M., Wang, Z., Jian, L.,... Liao, Q. (2024). Progesterone Enhances Niraparib Efficacy in Ovarian Cancer by Promoting Palmitoleic-Acid-Mediated Ferroptosis. Research, 7, 0371. doi: 10.34133/research.0371

Jiang, C., Sun, T., Xiang, D., Wei, S., & Li, W. (2018). Anticancer activity and mechanism of xanthohumol: a prenylated flavonoid from hops (Humulus lupulus L.). Frontiers in pharmacology, 9, 530. doi: https://doi.org/10.3389/fphar.2018.00530

Lin, X., Liao, Y., Chen, X., Long, D., Yu, T.,... Shen, F. (2016). Regulation of Oncoprotein 18/Stathmin Signaling by ERK Concerns the Resistance to Taxol in Nonsmall Cell Lung Cancer Cells. CancerBiotherapy and Radiopharmaceuticals, 31(2), 37-43. doi: 10.1089/cbr.2015.1921

Lin, X., Yu, T., Zhang, L., Chen, S., Chen, X., Liao, Y.,... Shen, F. (2016). Silencing Op18/stathmin by RNA Interference Promotes the Sensitivity of Nasopharyngeal Carcinoma Cells to Taxol and High-Grade Differentiation of Xenografted Tumours in Nude Mice. Basic & Clinical Pharmacology & Toxicology,19(6), 611-620. doi: https://doi.org/10.1111/bcpt.12633

Sun, D., Li, X., Nie, S., Liu, J., & Wang, S. (2023). Disorders of cancer metabolism: The therapeutic potential of cannabinoids. Biomedicine & Pharmacotherapy, 157, 113993. doi: https://doi.org/10.1016/j.biopha.2022.113993

Li, H., Jiang, Y., Wang, Y., Lv, H., Xie, H., Yang, G.,... Tang, T. (2018). The Effects of Warfarin on the Pharmacokinetics of Senkyunolide I in a Rat Model of Biliary Drainage After Administration of Chuanxiong. Frontiers in Pharmacology, 9, 1461. doi: 10.3389/fphar.2018.01461

Srikanth, D.; Shanthi, K.; Paoletti, N.; Joshi, S. V.; Shaik, M. G.; Rana, P.; Vadakattu, M.; Yaddanapudi, V. M.; Supuran, C. T.; Nanduri, S. Exploration of 1,3,5-Trisubstituted Pyrazoline Derivatives as Human Carbonic Anhydrase Inhibitors: Synthesis, Biological Evaluation and in Silico Studies. Int. J. Biol. Macromol. 2024, 280, 135890. https://doi.org/10.1016/j.ijbiomac.2024.135890

Zhou, J., Li, H., Wu, B., Zhu, L., Huang, Q., Guo, Z.,... Guo, T. (2024). Network pharmacology combined with experimental verification to explore the potential mechanism of naringenin in the treatment of cervical cancer. Scientific Reports, 14(1), 1860. doi: 10.1038/s41598-024-52413-9

Lou, Y., Song, F., Cheng, M., Hu, Y., Chai, Y., Hu, Q.,... Zhang, Y. (2023). Effects of the CYP3A inhibitors, voriconazole, itraconazole, and fluconazole on the pharmacokinetics of osimertinib in rats. PeerJ, 11, e15844. doi: https://doi.org/10.7717/peerj.15844

Cao, Z., Zhu, J., Wang, Z., Peng, Y., & Zeng, L. (2024). Comprehensive pan-cancer analysis reveals ENC1 as a promising prognostic biomarker for tumor microenvironment and therapeutic responses. Scientific Reports, 14(1), 25331. doi: 10.1038/s41598-024-76798-9

Wu, B., Wang, Z., Xie, H., & Xie, P. (2025). Dimethyl Fumarate Augments Anticancer Activity of Ångstrom Silver Particles in Myeloma Cells through NRF2 Activation. Advanced Therapeutics, 8(1), 2400363. doi: https://doi.org/10.1002/adtp.202400363

Pavitra S. Thacker, Prerna L. Tiwari Andrea Angeli, D. Srikanth, Baijayantimala Swain, Mohammed Arifuddin and Claudiu T. Supuran. Synthesis and biological evaluation of coumarin-linked 1,2,3-triazoles as potent inhibitors of carbonic anhydrases IX and XIII involved in tumorigenesis, MDPI,2021.

Qi, Y., Zheng, J., & Liu, L. (2024). Mirror-image protein and peptide drug discovery through mirror-image phage display. Chem, 10(8), 2390-2407. doi: 10.1016/j.chempr.2024.06.004

Fu, X., Yin, H., Chen, X., Yao, J., Ma, Y., Song, M.,... Wang, K. (2024). Three Rounds of Stability-Guided Optimization and Systematical Evaluation of Oncolytic Peptide LTX-315. Journal of Medicinal Chemistry, 67(5), 3885-3908. doi: 10.1021/acs.jmedchem.3c02232

6. PLOS authors have the option to publish the peer review history of their article (what does this mean? ). If published, this will include your full peer review and any attached files.

**Do you want your identity to be public for this peer review?** For information about this choice, including consent withdrawal, please see our Privacy Policy .

Reviewer #1: No

Reviewer #2: **Yes: ** Danaboina Srikanth

---

## [Author Response · Author response to Decision Letter 0]

17 Apr 2025

All of our responses to reviewers are contained in our submitted Word document titled, "Response-to-Reviewers.docx."

---

## [Editor Report · Decision Letter 1]

22 Apr 2025

Improved drug-screening tests of candidate anti-cancer drugs in patient-derived xenografts through use of numerous measures of tumor growth determined in multiple independent laboratories

PONE-D-24-51054R1

Dear Dr. Gordon,

We’re pleased to inform you that your manuscript has been judged scientifically suitable for publication and will be formally accepted for publication once it meets all outstanding technical requirements.

Kind regards,

Afzal Basha Shaik, Ph.D

Academic Editor

PLOS ONE

Additional Editor Comments (optional):

The manuscript is much improved according to the reviewers comments and is now suitable for publication.
---

## [Editor Report · Acceptance letter]

PONE-D-24-51054R1

PLOS ONE

Dear Dr. Gordon,

I'm pleased to inform you that your manuscript has been deemed suitable for publication in PLOS ONE. Congratulations! Your manuscript is now being handed over to our production team.

Kind regards,

on behalf of

Dr. Afzal Basha Shaik

Academic Editor

PLOS ONE